# Stereodivergent synthesis of vicinal quaternary-quaternary stereocenters and bioactive hyperolactones

Haifeng Zheng[1], Yan Wang[1], Chaoran Xu[1], Xi Xu[1], Lili Lin[1], Xiaohua Liu[1] & Xiaoming Feng[1,2]

Although great success has been achieved in asymmetric Claisen rearrangement for the synthesis of chiral γ,δ-unsaturated carbonyl compounds bearing vicinal tertiary-quaternary stereocenters, the development of asymmetric versions for stereodivergent construction of adjacent quaternary-quaternary stereocenters remains a formidable challenge because of the high steric hindrance. Here we report a catalytic enantioselective dearomatization Claisen rearrangement of allyl furyl ethers catalyzed by chiral *N,N*'-dioxide-Ni[II] complex catalysts. A variety of chiral γ,δ-unsaturated carbonyl compounds bearing vicinal quaternary-quaternary stereocenters were obtained with excellent outcomes under mild conditions. Furthermore, we disclosed that by matching the configuration of the catalysts and the alkene unit of the substrates, four stereoisomers of the products could be prepared in excellent yields and stereoselectivities. Finally, the fascination of this strategy was demonstrated by stereo-divergent synthesis of bioactive natural products hyperolactones B, C, and their epimers. A possible catalytic model was proposed to explain the origin of the asymmetric induction.

---

[1] Key Laboratory of Green Chemistry & Technology, Ministry of Education, College of Chemistry, Sichuan University, Chengdu 610064, China. [2] Collaborative Innovation Center of Chemical Science and Engineering, Tianjin 300071, China. Correspondence and requests for materials should be addressed to X.L. (email: liuxh@scu.edu.cn) or to X.F. (email: xmfeng@scu.edu.cn)

The Claisen rearrangement of allyl vinyl ethers is one of the most reliable and useful methods for synthesizing γ,δ-unsaturated carbonyl compounds, valuable intermediates in natural product construction[1–4]. On the basis of chiral Lewis acids[5–10], Jacobsen's guanidinium salts[11], N-heterocyclic carbenes[12], or transition metals[13–15] catalysts, various of linear and cyclic vinyl units have been transformed via catalytic asymmetric Claisen rearrangement to afford chiral γ,δ-unsaturated carbonyl compounds bearing vicinal tertiary-quaternary stereocenters (Fig. 1a). However, the direct and catalytic stereoselective assembly of these compounds with two adjacent quaternary-quaternary stereocenters remains scarce, because of the inherent steric congestion in the formation of C–C bond. To obtain vicinal quaternary-quaternary arrays and accomplish the total synthesis of several natural products, chiral starting materials were usually conducted through this type of reaction. Zhai and co-workers reported the total synthesis of (−)-Jiadifenin, in which the key intermediate was acquired on the basis of Ireland–Claisen rearrangement of chiral acetal precursor[16]. Nakamura developed a stereoselective Ireland–Claisen Rearrangement for the synthesis of the CDE ring system of antitumor and chiral building blocks for oxygenated Terpenoids[17,18]. In regard of catalytic asymmetric methodology, in 2010, the Jacobsen group reported the only catalytic example to produce the vicinal quaternary-quaternary array[19]. However, limited substrates were evaluated though good results were obtained, which restricted further application in the total synthesis of natural products. Although great achievement has been made in asymmetric Claisen rearrangement, the direct catalytic stereodivergent accessing to optically pure γ,δ-unsaturated carbonyl compounds bearing two adjacent quaternary stereocenters is yet to be described and would be of great value for the natural product-oriented asymmetric synthesis.

The hyperolactones A–C[20,21] and (−)-biyouyanagin A[22] are a family of spirolactone natural products isolated from *Hypericum chinese L*, which possess significant activity against HIV replication (Fig. 1b). These compounds contain a mutual spirolactone fragment with two chiral vicinal quaternary carbon centers. Besides, the configuration of hyperolactones A, C, and (−)-biyouyanagin A is (S,S), while the configuration of hyperolactone B is (R,S), which varies from the spirocyclic center. Because of their anti-HIV biological activities and unique structures, synthetic chemists have developed many methodologies for their total syntheses[22–32]. Particularly, Kraus utilized a tandem Claisen rearrangement to realize the synthesis of racemic hyperolactones C[33]. Though, the yield was only 25% even heating the reaction mixture in a sealed tube at 130 °C for 15 h, which indicated the challenge of the construction of adjacent quaternary stereocenters. Motivated by these initial results, we try to establish a general and efficient asymmetric Claisen rearrangement methodology for the synthesis of the family compounds of hyperolactone. Besides, we envision that by matching the configuration of the chiral catalysts and the substrates, all of the four stereoisomers might be accessible (Fig. 1c).

Herein, we report the application of chiral N,N′-dioxide-Ni[II] complex catalysts in the catalytic asymmetric dearomatization Claisen rearrangement of allyl furyl ethers. A series of γ,δ-unsaturated carbonyl compounds bearing two adjacent quaternary stereocenters were obtained in up to 99% yield, 19:1 dr, and 99% ee under mild reaction conditions. Moreover, the catalytic reaction shows a broad substrate scope, which may lead to the discovery of new bioactive molecules. Furthermore, four stereoisomers could be obtained and stereodivergent synthesis[34–48] of natural products hyperolactone B, C could be achieved as well.

## Results

**Optimization of reaction conditions**. Initially, we selected compound **1a** with *E*-configuration as the model substrate to optimize the reaction conditions. The classical catalysts of Cu(OTf)$_2$/**BOX** and Ni(OTf)$_2$/**BOX** yielded trace amount of the

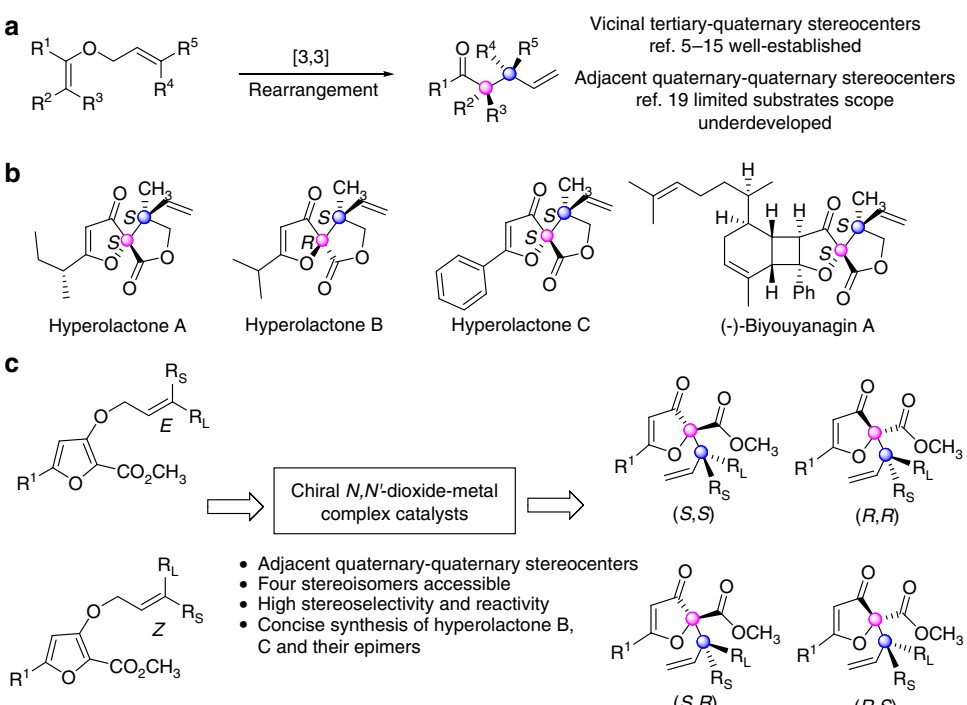

**Fig. 1** Asymmetric Claisen rearrangement of allyl vinyl ethers and representative natural products containing adjacent quaternary stereocenters. **a** Previous work for the catalytic asymmetric Claisen rearrangement of allyl vinyl ethers. **b** The structures of natural products hyperolactones A–C and (−)-biyouyanagin A. **c** Our catalytic asymmetric Claisen rearrangement strategy for stereodivergent construction of adjacent quaternary stereocenters

**Table 1 Optimization of the reaction conditions[a]**

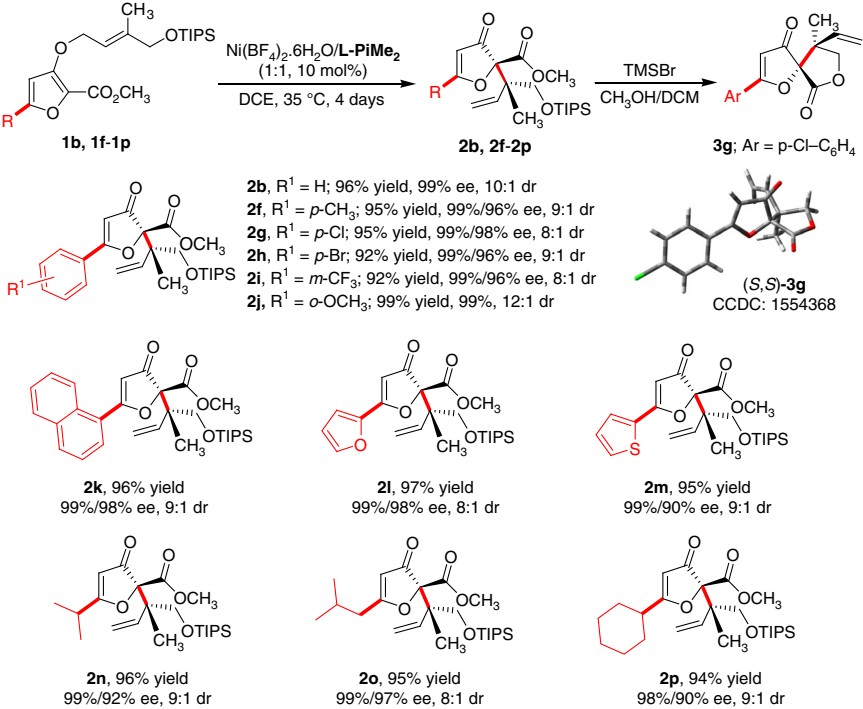

| Entry | 1 | Metal source/L | Yield (%)[b] | dr[c] | ee[d] |
|---|---|---|---|---|---|
| 1 | 1a | Cu(OTf)$_2$/**BOX** | trace | – | – |
| 2 | 1a | Ni(OTf)$_2$/**BOX** | trace | – | – |
| 3 | 1a | Yb(OTf)$_3$/**L-PiMe$_2$** | 47 | 3:1 | 33 |
| 4 | 1a | Ni(OTf)$_2$/**L-PiMe$_2$** | 85 | 5:1 | 98 |
| 5 | 1a | Ni(OTf)$_2$/**L-PiMe$_3$** | 85 | 5:1 | 98 |
| 6 | 1a | Ni(OTf)$_2$/**L-RaMe$_2$** | 90 | 2:1 | 94 |
| 7 | 1b | Ni(OTf)$_2$/**L-PiMe$_2$** | 90 | 8:1 | 98 |
| 8 | 1c | Ni(OTf)$_2$/**L-PiMe$_2$** | 90 | 5:1 | 98 |
| 9 | 1b | Ni(BF$_4$)$_2$·6H$_2$O/**L-PiMe$_2$** | 96 | 8:1 | 98 |
| 10[e] | 1b | Ni(BF$_4$)$_2$·6H$_2$O/**L-PiMe$_2$** | 96 | 10:1 | 99 |
| 11[e] | 1d | Ni(BF$_4$)$_2$·6H$_2$O/**L-PiMe$_2$** | 94 | 5:1 | 99 |
| 12[e] | 1e | Ni(BF$_4$)$_2$·6H$_2$O/**L-PiMe$_2$** | 90 | 1.5:1 | 99 |

[a]Unless otherwise noted, the reactions were performed with **1** (0.1 mmol), and metal source/**Ligand** (1:1, 10 mol%), in DCE (1.0 mL) at 70 °C for 4 h
[b]Isolated yield
[c]The dr value was determined by $^1$H NMR and HPLC analysis
[d]Determined by HPLC analysis on chiral stationary phases
[e]At 35 °C for 4 days

**Fig. 2** Substrate scope with variations at the furan unit. Unless otherwise noted, the reactions were performed with 1, Ni(BF$_4$)$_2$·6H$_2$O/L-PiMe$_2$ (1:1, 10 mol%), in DCE at 35 °C for 4 days. Isolated yield. The dr value was determined by $^1$H NMR and HPLC analysis. The ee value was determined by HPLC analysis on chiral stationary phases. The reaction of 2b and 2k were scaled up to 1.0 mmol. The reaction of 2n-2p were performed at 35 °C for 8 days

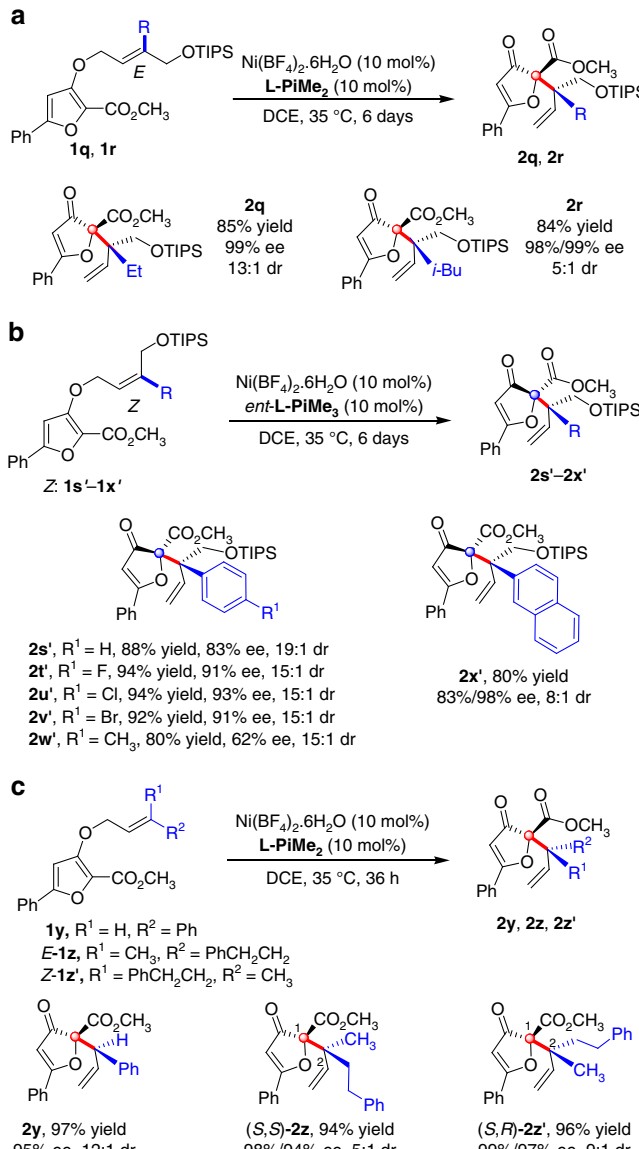

**Fig. 3** Substrate scope with variations at alkene unit. **a** Substrate scope of *E*-alkyl substituted alkene. **b** Substrate scope of *Z*-aryl substituted alkene; **c** Substrate scope of mono- and disubstituted alkene. The reactions were performed with 1, Ligand/Ni(BF$_4$)$_2$·6H$_2$O (1:1, 10 mol%), in DCE at 35 °C for indicated time. Isolated yield. The dr value was determined by $^1$H NMR and HPLC analysis. The ee value was determined by HPLC analysis on chiral stationary phases

the corresponding product **2b** in 90% yield, 8:1 dr with a comparable ee value (Table 1, entry 7). After variation of the metal counter ion, Ni(BF$_4$)$_2$·6H$_2$O was found to improve the yield to 96% (Table 1, entry 9). Furthermore, we discovered that the reaction temperature even could be reduced to 35 °C by prolonging the reaction time to 4 days and the desired product **2b** could be obtained in 96% yield, 10:1 dr and 99% ee (Table 1, entry 10). But further decreasing the reaction temperature led to low conversion. Variation of ester group on furyl unit of the substrate was also explored in the reaction. However, the yield and enantioselectivity were excellent but the dr was moderate (Table 1, entries 11 and 12).

**Substrate scope.** With the optimized reaction conditions identified, a wide range of substrates with varied substituents on furan unit were investigated. In all cases, the reaction proceeded smoothly with excellent yields and stereoselectivities (Fig. 2). Either electron-withdrawing (**1g**–**1i**) or electron-donating (**1f** and **1j**) substituents on the 5-phenyl ring of the furan unit were tolerated in the reaction, furnishing the corresponding products (**2f**–**2j**) in excellent yields, diastereo- and enantioselectivities (up to 99% yield, 12:1 dr and 99% ee). 1-Naphthyl (**1k**) and heteroaromatic (**1l** and **1m**) substituted furan substrates performed the reaction well, giving the related products (**2k**–**2m**) in 94–96% yields, 8:1–9:1 dr and 99% ee. Besides, when the 5-alkyl substituted furan substrates (**1n**–**1p**) were conducted in the reaction, the products (**2n**–**2p**) could also be obtained in excellent results (up to 96% yield, 9:1 dr and 99% ee) after prolonging the reaction time. To demonstrate the utility of this Claisen rearrangement, preparative-scale synthesis of the products **2b** and **2k** was carried out. Delightedly, the reactivity and stereoselectivity were maintained, which indicated that this method could tolerate the gram-scale chemical production. Furthermore, the absolute configuration of **2g** was determined to be (*S,S*) by the X-ray diffraction of the related spirolactone derivative **3g** (See Supplementary Fig. 76) (CCDC 1554368 (**3g**)). The compouds **2b** and **2n** exhibited a similar Cotton effect in their circular dichroism (CD) spectra (See Supplementary Figs. 64–75).

The substrates with variation at the alkene unit were also examined (Fig. 3). Under the optimized reaction conditions, *E*-ethyl substituted substrate **1q** and *E*-isobutyl substituted substrate **1r** were transformed into the desired products **2q** and **2r** in excellent yields and enantioselectivities (Fig. 3a, up to 85% yield and 99% ee). The latter showed relatively lower diastereoselectivity (**2q/2r**,13:1/5:1dr), which is probably due to the larger steric congestion that affects the stereoselective formation of C–C bond[16,19]. When *E*-phenyl substituted alkene substrate (*E*-**1s**) was treated in the reaction, no reaction was detected even raising the reaction temperature to 80 °C, which might be caused by the disfavored 1,3-repulsion interaction between aryl group and furan group in the reaction model (See Supplementary Fig. 80). Inversely, *Z*-aryl substituted alkene substrates could afford the corresponding diastereo-isomers **2′** as the major ones. We found the ligand **L-PiMe$_3$** was superior to **L-PiMe$_2$** in terms of reactivity and stereoselectivity, and the use of Ni(BF$_4$)$_2$/*ent*-**L-PiMe$_3$** catalyst for the reaction of *Z*-**1s′** resulted in better formation of the related isomer **2s′** compared with Ni(BF$_4$)$_2$/**L-PiMe$_2$** catalyst (89% yield, 19:1 dr, 83% ee vs. 75% yield, 9:1 dr, 80% ee). With Ni(BF$_4$)$_2$/*ent*-**L-PiMe$_3$** catalyst, the substrates *Z*-**1s′** to *Z*-**1x′** could participate in the Claisen rearrangement to afford the desired products **2s′**–**2x′** in excellent yields, stereoselectivities (Fig. 3b, up to 95% yield, 19:1 dr and 93% ee). Halogen substitution on the phenyl ring (*Z*-**1t′**–*Z*-**1v′**) made a significant increase of the enantioselectivity. However, electron-donating functional group on the phenyl ring (*Z*-**1w′**) decreased the enantioselectivity to

targeted product **2a** even at 70 °C (Table 1, entries 1 and 2)[5]. Next, a range of Lewis acids were examined by complexing with *N,N*′-dioxide ligand (**L-PiMe$_2$**) derived from *L*-pipecolic acid in DCE at 70 °C (For details, see Supplementary Table 1). The reaction proceeded sluggishly in the presence of Yb(OTf)$_3$ (Table 1, entriy 3). Fortunately, Ni(OTf)$_2$ showed excellent catalytic ability and the reaction could complete within 4 h to afford the desired product **2a** in 85% yield with 5:1 dr and 98% ee (Table 1, entry 4). Inspired by these results, other *N,N*′-dioxide ligands were evaluated, however, no better results were obtained (Table 1, entries 5 and 6). Our further attempts to improve the yield and diastereoselectivity focused on the investigation of different silyl protecting group of the substrate **1**. Substrates **1b** and **1c** were synthesized to inspect the reaction (entries 7 and 8). The substrate **1b** with bulky TIPS group resulted in the formation of the

**Fig. 4** Synthesis of all four stereoisomers. Unless otherwise noted, the reactions were performed with 1 (0.1 mmol), and cat* or *ent*-cat* (10 mol%), in DCE (1.0 mL) at 35 °C for 4 days. Isolated yield. The dr value was determined by $^1$H NMR and HPLC analysis. The ee value was determined by HPLC analysis on chiral stationary phases. cat* = Ni(BF$_4$)$_2$·6H$_2$O/L-PiMe$_2$ (1:1), *ent*-cat* = Ni(BF$_4$)$_2$·6H$_2$O/*ent*-L-PiMe$_2$ (1:1)

**Fig. 5** Stereodivergent synthesis of natural products hyperolactones B, C, (−)-biyouyanagin A, and their epimers. **a** The deprotection/lactonization transformation of the product **2b**, **2b′**, **2n** and **2n′**. **b** Formal synthesis of (−)-biyouyanagin A and *epi*-(−)-biyouyanagin A

62% ee. In addition, the 2-naphthyl substituent alkene *E*-**1x′** was found suitable for this reaction, delivering the product **2x′** in 82% yield with 8:1 dr and 83% ee. Furthermore, terminal monophenyl substituted alkene substrate *E*-**1y** could afford the desired product **2y** in 97% yield with 95% ee and 12:1 dr in the presence of Ni(BF$_4$)$_2$/**L-PiMe$_2$** catalyst. Both *E*-**1z** and *Z*-**1z′** with terminal disubstituent were suitable in this reaction to afford the corresponding products **2z** and **2z′** in excellent results (Fig. 3c, up to 96% yield, 99% ee and 9:1 dr).

**Asymmetric synthesis of four stereoisomers.** To examine our initial stereodivergent assumption and the purpose of synthesizing natural products hyperolactones B and C, we prepared the *E*-**1b**, **1n** and *Z*-**1b′**, **1n′** and treated them under the optimal reaction conditions (Fig. 4). As expected, the *E*-substrates preferably afforded (*S,S*)-configuration products **2b** and **2n**, while, the *Z*-substrates yielded the (*S,R*)-configuration products **2b′** and **2n′** (The configuration of the products were confirmed by CD spectra and NOE spectra analysis of hyperolactones, see

Supplementary Figs. 64–79). Besides, by alternating the absolute configuration of N,N′-dioxide ligand **L-PiMe$_2$**, the corresponding *ent*-products were available in comparable results. Thus, all four stereoisomers of such γ,δ-unsaturated carbonyl compounds containing vicinal quaternary-quaternary centers could be obtained with this kind of easily available chiral catalyst.

**Stereodivergent synthesis of hyperolactones.** Subsequently, we tried various reagents to realize the deprotection/lactonization transformation of the product **2b** and **2n**. The conditions, such as TBAF, PPTS, and hydrogen fluoride-pyridine solution proved unsuccessful (For details, see Supplementary Table 8). Delightfully, in the presence of TMSBr reagent, the optically active (*S,S*)-**2b**, (*S,R*)-**2b′**, (*R,S*)-**2n′** and (*S,S*)-**2n** could participate in the transformation well, delivering hyperolactones B, C and *epi*-hyperolactones B, C in excellent yields with the diastereo- and enantioselectivity maintained (For details, see Supplementary Note 11). Accordingly, the access to natural product (−)-biyouyanagin A and its epimer is possible from the key

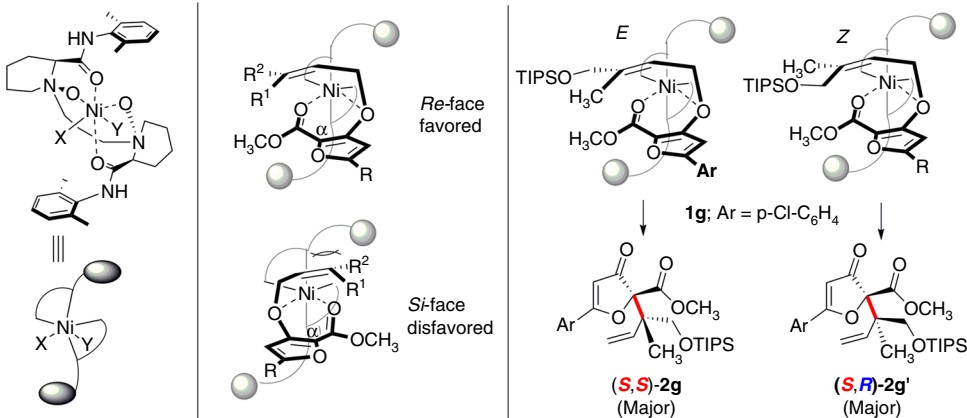

**Fig. 6** Proposed stereochemical model. Substrate **1g** was selected for the model analysis

reaction intermediates[26,27,29,31] of hyperolactone C or *epi*-hyperolactone C (Fig. 5).

**Proposed stereochemical model**. On the basis of our previous works[49–51] and the absolute configuration of the product **2g**, a possible stereocontrol model was proposed in Fig. 6. The chiral *N,N′*-dioxide coordinates to the nickel(II) center to form octahedral metal complex in which the chiral ligand occupies four sites. The substrate **1g** displays the ancillary solvents or anion and could coordinate to the Lewis acid catalyst via a bidentate manner with the oxygen of ether and carbonyl group of the auxiliary ester group. Due to the steric hindrance between alkene unit and the bulky aniline moiety of ligand, the alkene preferentially approaches α position of furan unit from the *Re* face. Therefore, the *E*-alkene substrate **1g** affords (*S,S*)-**2g** and the *Z*-alkene substrate **1g′** affords (*S,R*)-**2g′**.

## Discussion

In summary, we have developed a catalytic asymmetric dearomatization Claisen rearrangement of allyl furyl ethers for the synthesis of chiral γ,δ-unsaturated carbonyl compounds with two adjacent quaternary-quaternary carbon centers. By adjusting the configuration of the catalysts and alkene units, all four possible product stereoisomers could be prepared in excellent yields and stereoselectivities. In addition, the reaction has a broad substrate scope, which may lead to the discovery of new bioactive molecules. Furthermore, the bioactive natural products hyperolactones B, C, and their epimers could be readily acquired in high efficiency, which are also the key intermediates for the synthesis of natural product (−)-biyouyanagin A and its epimer.

## Methods

**General procedure for asymmetric Claisen rearrangement**. *N,N′*-Dioxide ligand **L-PiMe₂** or *ent*-**L-PiMe₂** or *ent*-**L-PiMe₃** (10 mol%), Ni(BF₄)₂·6H₂O (10 mol%) in CH₂Cl₂ were stirred at 35 °C for 1 h. Then CH₂Cl₂ was removed in *vacuo*. Substrate **1** and DCE were added and the resulting mixture was stirred at 35 °C for 4–8 days. After the reaction completed (monitored by TLC), flash column chromatography was carried out to provide the desired product (Petroleum ether/EtOAc = 20:1 to 10:1).

**General procedure for deprotection/cyclization reaction**. To a solution of the product **2** (0.1 mmol) in CH₃OH/CH₂Cl₂ (1.5 mL, 2:1, v/v), TMSBr (0.4 mmol, 4.0 equiv.) was added, the reaction mixture was stirred at 30 °C for 8–24 h. Then reaction was quenched by saturated aq. Na₂CO₃ (1 mL) and diluted with water (1 mL). The product was extracted with ethyl acetate (3 × 10 mL). The combined organic extracts were washed with brine (20 mL) and concentrated in vacuo. The product was purified by column chromatography on silica gel (hexane/EtOAc = 10/1 as eluent).

**Data availability**. The X-ray crystallographic coordinates for structures reported in this article have been deposited at the Cambridge Crystallographic Data Centre (CCDC). Crystal data and structure refinement for **3g** was displayed in Supplementary Information. These data can be obtained free of charge from The Cambridge Crystallographic Data Centre via https://www.ccdc.cam.ac.uk/ under deposition number 1554368. All other data is available from the corresponding author upon reasonable request.

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

## Acknowledgements

We appreciate the National Natural Science Foundation of China (Nos. 21432006, 21625205) for financial support. Dedicated to professor George A. Kraus for his initial contribution on the dearomatization Claisen rearrangement of allyl furyl ethers.

## Author contributions

H.F.Z. performed experiments and prepared the Supplementary Information and Manuscript. Y.W. and C.R.X. took part in the reaction development and synthesized several substrates. X.X. repeated some experiments. L.L.L. helped with modifying the paper and Supplementary Information. X.H.L. and X.M.F. conceived and directed the project.

## Additional information

**Competing interests:** The authors declare no competing interests.

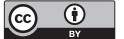

