## [Peer Review File · Nature Communications]

Reviewers' comments:

Reviewer #1 (Remarks to the Author):

In this manuscript, Feng, Liu and co-workers reported a novel enantioselective dearomatization Claisen rearrangement of allyl furyl ethers catalyzed by chiral N,N'-dioxide-Ni(II) complex catalysts, which provides a rapid access of various chiral γ,δ -unsaturated carbonyl compounds bearing vicinal tertiary quaternary stereocenters. Importantly, the resulting products represent a key structural element in numerous bioactive natural products, and thus the development of synthetic methods for their syntheses is of great importance. As mentioned by the authors, although a number of asymmetric Claisen rearrangement have been established to access the chiral γ,δ -unsaturated carbonyl compounds bearing a single quaternary carbon, the method for the direct assembly of the products with two adjacent quaternary stereocenters remains scarce, mainly due to the inherent steric effect. In this context, the present work provides a powerful and practical solution to this challenge. Key to their success relies on the discovery of a unique catalytic system, the combination of chiral N,N'-dioxide ligand (developed by their own group) and Ni(BF₄)₂·6H₂O. The reported method displays a broad substrate scope, excellent yield and enantioselectivity. More impressively, the method enables the stereodivergent access of four possible diastereoisomers by judicious selection of the suitable catalyst and substrate. Finally, as proof-of-concept case, the newly developed method has been applied to the asymmetric syntheses of hyperolactones B, C and their epimers. Taken all, the developed chemistry represents a significant improvement in the field of catalytic asymmetric Claisen rearrangement. The work is well implemented, and the supplementary data appears solid. I believe this work would arise extensive readership from synthetic community. Also, the synthetic method would find broad utility in organic synthesis, particularly in the natural product synthesis. Taken all, I recommend its acceptance by Nature Communication with minor revisions addressed as follow:

- 1) The title of the manuscript, "a Concise Enantioselective Synthesis of Anti-HIV Agents Hyperolactones B, C and their Epimers", is misleading. In the work, the authors only achieve the enantioselective synthesis of the natural products hyperolactones B, C and their Epimers. However, no anti-HCV reactivity reported for these natural product in the isolation paper. Instead, it was (-)-biyouyanagin A and its analogs that have been reported as anti-HCV agents. In this work, only the formal synthesis of biyouyanagin A was described.
- 2) It was found that Ni(BF₄)₂·6H₂O was the optimal choice of catalyst. Is the "H₂O" play some role in this reaction? The author might conduct a control reaction with Ni(BF₄)₂ if it is commercially available.
- 3) While the absolute configuration of the products were determined by the combination of the CD spectrum and NOE spectrum, a direct comparison of the specific rotation of the synthetic hyperolactones B and C to those reported for natural products (or the data in other synthetic work) should be provided in the work, since it will afford the more direct and convincing evidence for their structural determination.
- 4) An Asymmetric Claisen rearrangement of allyl vinyl ethers has been reported by Tang and co-workers (Angew. Chem. Int. Ed. 2013, 52, 4198 –4202), which affords the chiral dihydrofuran-3-ones similar to the present work. This work should be included in the reference 5-9.
- 5) A number of typo and grammar errors should be correct, some of which are listed below:
 - a) Abstract, line 12, "condition" should be revised to "conditions"
 - b) Page 1, line 33, "albeit" should be revised to "though".
 - c) Page 1, line 34, "restrict" should be correct to "restricts".
 - d) Page 1, Scheme 1A, it is better to change "adjacent quaternary quaternary stereocenters" to "adjacent quaternary-quaternary stereocenters", so does the "vicinal tertiary quaternary stereocenters". The current form is misleading.
 - e) Page 2, line 47, "many total synthesis" should be "many total syntheses".
 - f) Page 4, line 113, "relative low" should be "relatively low".
 - g) Page 5, line 119, "L-PiMe₃ was prior to L-PiMe₂" should be "L-PiMe₃ was superior to L-PiMe₂".
 - h) Page 5, line 161, "epimeris" should be "epimers".

There are a lot of other errors which are not pointed out, and the author should check their

manuscript carefully to improve their manuscript before its publication.

Reviewer #2 (Remarks to the Author):

In this manuscript, Liu and Feng report an enantioselective Claisen rearrangement of furan-derived substrates and an application of this method to the asymmetric synthesis of hyperlactones. Overall, this work represents a nice addition to the literature on asymmetric Claisen methodology. The closest related works are the papers from Hiersemann, which described the Cu-catalyzed rearrangements of O-allylated alpha-ketoesters (ref 5). The key innovation of this work is the extension of this approach to furan-derived substrates. A new catalyst is also described (a Ni complex of a bis-nitroxide donor as opposed to Cu(BOX) complexes). In my view, this study represents a significant synthetic advance, and there is a clear demonstration of utility in the synthesis of hyperlactone natural products. I think this would be an excellent addition to Nature Commun., and I can recommend acceptance with some minor questions/suggestions to the authors:

1) Given that the Cu(BOX) catalysts are the state-of-the-art for this reaction. I would be interested to see a comparison of these catalysts included in Table 1. Does the Ni(PiMe) catalyst offer a specific advantage? Or is it simply an alternative catalyst?

2) In table 1, I think it should be specified that the ee values are for the major diastereomer. Given that the dr values are <10:1, I also think that the ee of the minor could be included.

3) While I don't think this needs to be changed, I am curious why the authors elected to use the lower temperature and several days reaction time? It appears from Table 1 that the 70 oC conditions give quite adequate ee and dr values. Is there a steeper drop-off for other substrates?

The supporting information appears to be complete, with NMR data, HRMS, and HPLC traces for all new compounds. I also appreciate the inclusion of CD data to corroborate the absolute configuration assignments.

For reviewer 1:

Question: The title of the manuscript, “a Concise Enantioselective Synthesis of Anti-HIV Agents Hyperolactones B, C and their Epimers”, is misleading. In the work, the authors only achieve the enantioselective synthesis of the natural products hyperolactones B, C and their Epimers.

However, no anti-HCV reactivity reported for these natural product in the isolation paper.

Instead, it was (-)-biyouyanagin A and its analogs that have been reported as anti-HCV agents. In this work, only the formal synthesis of biyouyanagin A was described.

Reply: For this opinion, thanks for your careful deliberation. We have changed the title of the manuscript as “Stereodivergent Synthesis of Vicinal Quaternary-Quaternary Stereocenters: a Concise Enantioselective Synthesis of Natural Products Hyperolactones B, C and their Epimers”.

Question: It was found that $\text{Ni}(\text{BF}_4)_2 \cdot 6\text{H}_2\text{O}$ was the optimal choice of catalyst. Is the “ H_2O ” play some role in this reaction? The author might conduct a control reaction with $\text{Ni}(\text{BF}_4)_2$ if it is commercially available.

Reply: Thank you very much. It's a very good question. $\text{Ni}(\text{BF}_4)_2$ is not commercially available. We found that it is unstable at higher temperature, which decomposed at 122 °C during removal of H_2O from $\text{Ni}(\text{BF}_4)_2 \cdot 6\text{H}_2\text{O}$.

Alternatively, we did comparative experiments. *N,N'*-dioxide ligand (**L-PiMe₂**) (0.01 mmol) and $\text{Ni}(\text{BF}_4)_2 \cdot 6\text{H}_2\text{O}$ (0.01 mmol) in CH_2Cl_2 (1.0 mL) were stirred at 35 °C for 1h. Then removing CH_2Cl_2 and heating at 70 °C for 2h under a vacuum, affording the complex catalyst $\text{Ni}(\text{BF}_4)_2/\text{L-PiMe}_2$ as light green powder. With this catalyst in hand, we did two experiments. One was to use the catalyst directly, and the other was to add H_2O (5 uL/ 270 mol%) to the catalytic system. Both experiments yielded the same results as the original (equation 1 vs 2 vs 3). From these results, we could conclude that the “ H_2O ” did not play some role in this reaction and this reaction was not sensitive to a small amount of water.

Question: While the absolute configuration of the products were determined by the combination of the CD spectrum and NOE spectrum, a direct comparison of the specific rotation of the synthetic hyperolactones B and C to those reported for natural products (or the data in other synthetic work) should be provided in the work, since it will afford the more direct and convincing evidence for their structural determination.

Reply: For this question, we have did the experiments and the results were consistent with the literature. The rotation data have been added in the supporting information in detail.

Question: An asymmetric Claisen rearrangement of allyl vinyl ethers has been reported by Tang and co-workers (*Angew. Chem. Int. Ed.* **2013**, 52, 4198–4202), which affords the chiral dihydrofuran-3-ones similar to the present work. This work should be included in the reference.

Reply: We have added this work (*Angew. Chem. Int. Ed.* **2013**, 52, 4198–4202) in the reference 10.

Question: A number of typo and grammar errors should be correct, some of which are listed below:

- Abstract, line 12, “condition” should be revised to “conditions”
- Page 1, line 33, “albeit” should be revised to “though”.
- Page 1, line 34, “restrict” should be correct to “restricts”.
- Page 1, Scheme 1A, it is better to change “adjacent quaternary quaternary stereocenters” to “adjacent quaternary-quaternary stereocenters”, so does the “vicinal tertiary quaternary stereocenters”. The current form is misleading.
- Page 2, line 47, “many total synthesis” should be “many total syntheses”.
- Page 4, line 113, “relative low” should be “relatively low”.
- Page 5, line 119, “L-PiMe₃ was prior to L-PiMe₂” should be “L-PiMe₃ was superior to L-PiMe₂”.
- Page 5, line 161, “epimeris” should be “epimers”.

There are a lot of other errors which are not pointed out, and the author should check their manuscript carefully to improve their manuscript before its publication.

Reply: We are very grateful for this suggestion, it is very important. We have tried our best to modify the typo and grammar errors, the changes have been highlighted in the manuscript.

For reviewer 2:

Question: Given that the Cu(BOX) catalysts are the state-of-the-art for this reaction. I would be interested to see a comparison of these catalysts included in Table 1. Does the Ni(PiMe) catalyst offer a specific advantage? Or is it simply an alternative catalyst?

Reply: For this question, we synthesized two **BOX** ligands (**BOX-^tBu** and **BOX-Bn**) and conducted them in the model reaction with metal Cu(OTf)₂. No reaction occurred at 35 °C and trace amount of the product **2a** detected and partial material decomposed at 70 °C. In addition, Ni^{II}/(**BOX-^tBu**) catalyst was also used in the model reaction, however, trace amount of the product **2a** detected at 70 °C. These results have indicated the specific advantage for Ni^{II}(**PiMe**) catalyst. We have added two experiments [Cu^{II}(**BOX-^tBu**) and Ni^{II}/(**BOX-^tBu**) at 70 °C] in the Table 1 and supporting information, the corresponding description was also added in the manuscript.

Question: In table 1, I think it should be specified that the ee values are for the major diastereomer. Given that the dr values are <10:1, I also think that the ee of the minor could be included.

Reply: The ee of the minor products have been added in the manuscript and supporting information.

Question: While I don't think this needs to be changed, I am curious why the authors elected to use the lower temperature and several days reaction time? It appears from Table 1 that the 70 °C conditions give quite adequate ee and dr values. Is there a steeper drop-off for other substrates?

Reply: On the basis of this mind, we did several experiments. However, as follow: the substrates **E-1n**, **E-1z**, **Z-1z** were conducted at 70 °C conditions, the dr were steeper drop-off. These results indicated the challenge for controlling stereoselectivity at high reaction temperature. A larger energy difference is needed between the diastereomeric transition states at high temperatures to achieve the same level of selectivity at rt.

We hope that the revised version is satisfactory for publication in *Nature Communications*,
Thanks again for your assistance with the manuscript.

Best regards and look forward to hearing from you.

Sincerely yours,

Prof. Dr. Xiaoming Feng

REVIEWERS' COMMENTS:

Reviewer #1 (Remarks to the Author):

All of the questions raised by the reviewers have been addressed properly in the revised manuscript. Thus, I recommend its acceptance by Nature Communications without reservation.

Reviewer #2 (Remarks to the Author):

Liu and Feng have submitted a revised version of their manuscript on the asymmetric Claisen rearrangement reaction. In my initial review, I raised a few minor points regarding the inclusion of dr values for minor diastereomers, the choice of reaction temperature, and alternative BOX catalysts. All of these comments have been adequately addressed, and I recommend acceptance with no further changes.